# Laboratory Tests to Optimize the Milking Machine Settings with Air Inlet Teat Cups for Sheep and Goats

**Shehadeh Kaskous**

Department of Research and Development, Siliconform, Schelmengriesstrasse 1, 86842 Türkheim, Germany; skaskous@siliconform.com

**Abstract:** Milking machine design and performance are directly related to the milkability of sheep and goats, with the aim of milking quickly, completely and gently. This leads to an increase in the milk yield with improved quality, and the maintenance of healthy udders. The aim of this study was to carry out laboratory tests to determine the optimal level of vacuum, pulsation rate and pulsation ratio of new milking machines in high and low milk lines for sheep and goats. This study was conducted at the Department of Research and Development, Siliconform, Germany. For this purpose, different levels of vacuum (32, 34, 36, 38 and 40 kPa), milk jet (2, 2.5, 3 and 4 mm), milk line (high line and low line) and pulsation ratio (50:50 and 60:40) were used. First minute water flow (1st WF/kg) was used as an indicator for assessing the best combination in the milking machine. In addition, the cyclic vacuum fluctuation was measured in the inner chamber of the teat cup during the 1st WF/kg with the aid of a Vacuscope device. Statistical analysis was conducted using the mixed procedure in SAS. Our results show that the vacuum level, the milk jet and the pulsation ratio had a significant influence ($p < 0.05$) on the 1st WF/kg in the two milking machines for goats and sheep. In conclusion, the ideal conditions for milking goats with air inlet teat cups in the milking machine are a vacuum level of 36–38 kPa (low line) and 38–40 kPa (high line), a pulsation rate of 90 cycles/min and a pulsation ratio of 60:40, while the ideal conditions in the sheep milking machines are a vacuum level of 35–36 kPa (low line) and 36–38 kPa (high line), a pulsation rate of 120 cycles/min and a pulsation ratio of 60:40 or 50:50.

**Keywords:** goat; milking machine; pulsation rate; pulsation ratio; sheep; Siliconform; vacuum level



## 1. Introduction

In many countries around the world, especially in Asia, Africa and Mediterranean regions, dairy farming of sheep and goats is economically very important. Sheep and goat milk is a valuable raw material for different types of dairy products. In order to increase milk yield and achieve better milk quality from sheep and goats, a suitable milking machine with optimal settings must be used. An indicator of the appropriate sheep and goat milking machine is a low value of the somatic cell count (SCC) in the milk produced. This means that the udder remains healthy and the milking machine meets the physiological requirements of sheep and goats. Therefore, many authors use the SCC in the bulk tank as the main indicator of udder health in sheep and goats [1–5]. These cells can be strongly influenced by the milking system used [5] and the properties of the milking machine such as the vacuum level and pulsation rate and ratio [6–8]. On the other hand, the properties of the teat liner also play a major role in the health of the teats in sheep and goats, particularly the influence of the teat liner on the teat wall thickness, teat wall area and teat canal length during the milking process [9]. It is also necessary to use liners adapted to the morphological features of each type of teat (for sheep or goats), taking special care to check the liners regularly to ensure the benefits of the pulsation at the teat end. In addition, the daily milk yield and its quality in sheep and goats are influenced by the frequency with which the milking machine is used [10–12]. In general, the milkability of sheep and goats depends on the

animal physiology and the teat and udder morphology [13,14], the milking system and the structure of a milking machine [5,15,16] and farmers' experience with milking [17,18].

In the past, goat and sheep milking machines adapted for dairy cattle have been used, resulting in poor milk quality and udder health problems [19,20]. In addition, high vacuum levels (>42 kPa) were often used in the milking machine in order to facilitate the opening of the teat canal by overcoming the biological closing forces in the teat sphincter. However, this can lead to severe machine-related damage to the teat tissue [21]. In recent years, many sheep and goat milking machines have been developed. However, many farmers still suffer from the inadequate efficiency of these milking machines because they do not empty the udder completely and many udders become diseased (mastitis) [5,15,20,22,23]. The cause of this problem is the lack of a suitable milking machine adapted to the physiology of the udders of sheep and goats. Studies on Saanen goats in Sao Paulo have shown that if the correct milking machine meeting the udder requirement is not used, negative effects on udder properties and poor milk quality can be found [24]. Katanos et al. [15] emphasized that the best machine milking technique is routine milking without applying hand stripping in dairy goats. For this reason, Siliconform has developed milking machines for sheep and goats that include teat cups with an integrated air inlet. The company has over 50 years of experience in this field with many different species, such as cows, camels and horses. Evidence of this can be found in the latest publications from the Department of Research and Development of Siliconform, Türkheim, Germany [25–27].

Our hypothesis was to develop a new milking machine for sheep and goats that would guarantee high milk yield and quality while maintaining udder health. These new milking machines need to be tested in the laboratory and in the field. The aim of this study was therefore to carry out laboratory tests in order to determine the optimal level of vacuum, pulsation rate and pulsation ratio of the new milking machines for sheep and goats. This study is part of a project entitled "Development of milking machine for sheep and goats". After this phase, it is planned to carry out field tests in order to compare laboratory and field results and to implement the necessary measures. Finally, we will bring an optimal milking machine with the best features for sheep and goats to market.

## 2. Material and Methods

### 2.1. Experimental Design

The daily milking of sheep and goats should be quick, complete and gentle. These three basic requirements go hand in hand with the aim of farmers to use resources efficiently, to operate profitably and to continuously improve the general welfare and health of dairy goats and sheep. In order to meet and balance these three requirements, a suitable milking machine and a good udder are of great importance. A new concept has therefore been developed to identify factors influencing the mechanical milking process. Our investigations were based on two main areas: The first area (the milking system) comprises a number of parameters of the milking machine, all of which have an influence on successful milk removal. The second area (the animals) covers different teat diameters (milk jet), which also influence the milking performance. In practice, there are two other areas: the milking operator and the milking parlour. Scheme 1 presents an overview of the two areas of the milking process in laboratory tests.

- The milking system used is a milking unit with the following parts: two two-chamber teat cups, each with a short milk-and-pulse line, as well as a claw and a long milk-and-pulse line. Each teat cup has an air inlet and is divided into two spaces: the pulse chamber and the inner chamber. Many parameters can be evaluated in this area. We focused on the vacuum level, pulsation rate, pulsation ratio and height of the milk line. All teat cups are equipped with silicone teat liners.
- The animals (dairy goat and sheep) are the second area and are represented here as the diameter of the milk jet, corresponding to the teat diameter of the udder of goats and sheep.

- The water flow in the first minute/kg (1st WF/kg) is displayed here, i.e., the amount of water received (milked) in kilograms in the first minute after teat attachment. The 1st WF/kg of the milking process was measured as a consistent indicator for evaluating the best combination in the milking machine.

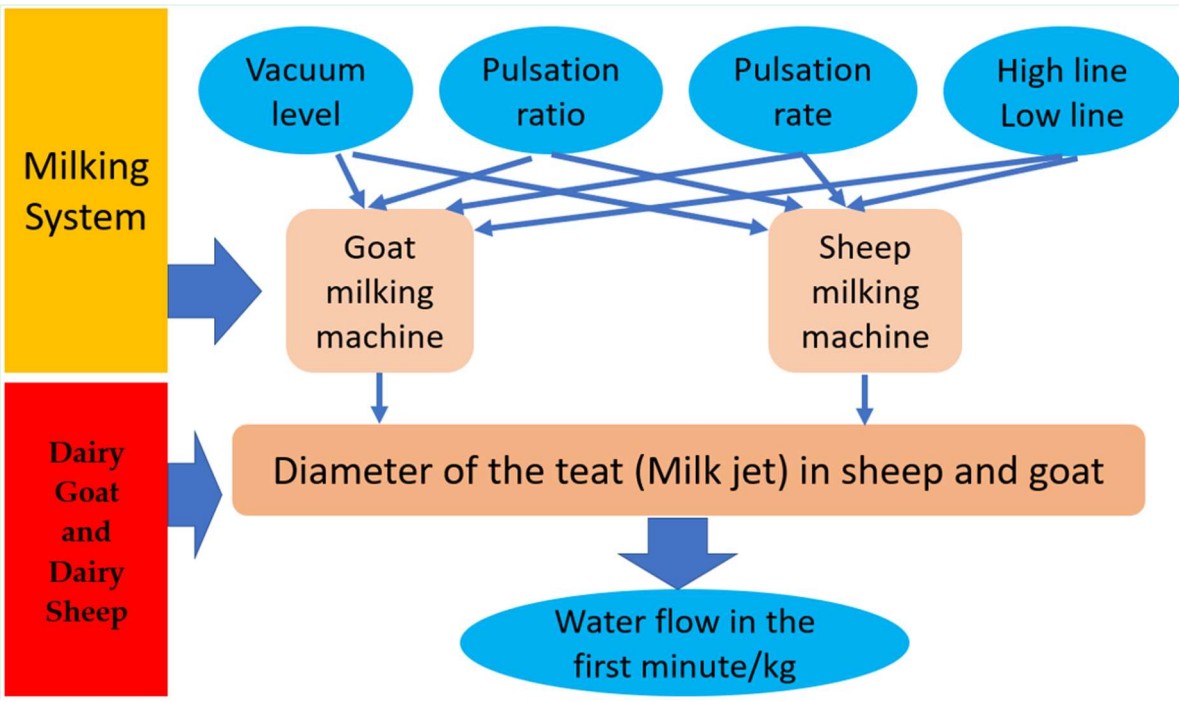

**Scheme 1.** Overview on methodology of the parameters of two areas: milking system and the animals (dairy goat and sheep).

The investigations were carried out in the laboratory at Siliconform, Türkheim, Germany.

### 2.2. Milking Laboratory

The milking laboratory used is equipped with a pipe milking system and has two milk pipelines (low line and high line) installed. In this system, all parameters to be tested can be easily altered. In addition, the milking laboratory is equipped with a Vacuscope device, which is used to measure the vacuum fluctuations in the inner and pulsation chambers of the teat cups.

### 2.3. Investigation Parameters

The following parameters were tested in the laboratory:

- Different diameters (2, 2.5, 3 and 4 mm) of milk jets were tested, which could be similar to the diameter of the streak canal of the teats in sheep and goats (ranging from 1.8 to 3.1 mm) [28];
- Multiple vacuum levels: 32, 34, 36, 38 and 40 kPa;
- Two pulsation rates: 90 cycles/min for the goat milking machine and 120 cycles/min for the sheep milking machine;
- Two pulsation ratios: 50:50 and 60:40 suction phase/rest phase (S/R);
- Two levels of the milk pipeline: high milk pipeline and low milk pipeline.

The examination parameters are shown in the following Scheme 2:

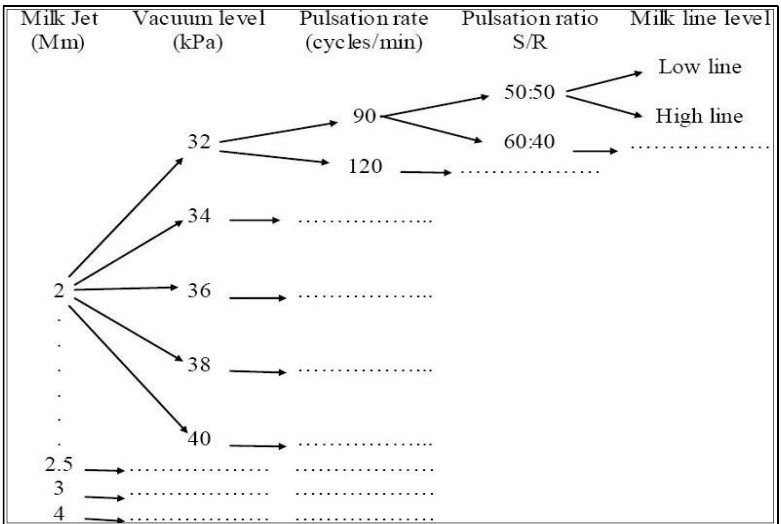

**Scheme 2.** Examination parameters used in the Siliconform laboratory.

*2.4. Milking Process*

The parameters tested (milk jet, vacuum level, pulsation rate and ratio) were set in the milking machine according to Scheme 2. The milking process was started and the 1st WF/kg recorded. At the same time, the cyclic vacuum fluctuations in the interior and pulsation chambers of the teat cup were recorded. The milking process was repeated 320 times; 160 times with the goat milking unit and 160 times with the sheep milking unit.

*2.5. Statistical Analyses*

Statistical analysis was performed using the mixed procedure in SAS [29]. The model included the fixed effects of milk jet, vacuum levels, pulsation rate, the level of the milk line (high or low) and pulsation ratio, and the random effect of repeats within the milking unit (sheep or goats):

$$Y_{ijklm} = \mu + J_i + VL_j + PR_k + PO_l + ML_m + e_{ijklm}$$

where $Y_{ijklm}$ = 1st WF/kg of the milking process during the investigation time of the $ijklm^{th}$ records;

$\mu$ = overall mean;

$J_i$ = fixed effects of $i^{th}$ milk jet (i = 2, 2.5, 3 and 4 mm);

$VL_j$ = fixed effects of $j^{th}$ vacuum level (j = 32, 34, 36, 38 and 40 kPa);

$PR_k$ = fixed effects of $k^{th}$ pulsation rate (k = 1 for goat milking machine, 2 for sheep milking machine);

$PO_l$ = fixed effects of $l^{th}$ pulsation ratio (l = 1 (50:50) and 2 (60:40));

$ML_m$ = fixed effects of $m^{th}$ height of the milk line (m = 1 (high line) and 2 (low line));

$e_{ijklm}$ = residual error.

The least squares mean was compared, and significance was declared at $p < 0.05$.

**3. Results and Discussion**

*3.1. Vacuum Levels, Pulsation Rates and Ratios and Height of the Milk Line for the Optimization of the Milking Machine for Sheep and Goats*

Table 1 lists the individual influencing factors with the corresponding degrees of freedom, the calculated F-values and the respective probability of exceedance (*p*-value). Statistical information for the milking machine for sheep and goats can be found in this table for the factors milk jet, vacuum level, pulsation ratio and height of the milk line.

**Table 1.** The individual effects with the associated degrees of freedom (Num DF/Den DF, the *F*-values and the *p*-values).

| Effect | Num DF | Den DF | *F*-Value | *p*-Value |
|---|---|---|---|---|
| | | Sheep milking machine | | |
| Milk jet | 3 | 139 | 64.18 | <0.0001 |
| Vacuum level | 4 | 139 | 15.95 | <0.0001 |
| Pulsation ratio | 1 | 139 | 1.82 | 0.1800 |
| Height of ML | 1 | 139 | 99.79 | <0.0001 |
| | | Goat milking machine | | |
| Milk jet | 3 | 116 | 61.63 | <0.0001 |
| Vacuum level | 4 | 116 | 5.34 | 0.0003 |
| Pulsation ratio | 1 | 116 | 1.9 | 0.7650 |
| Height of ML | 1 | 116 | 19.06 | <0.0001 |

Our results (see Table 2) show that the milk jet, the vacuum level and the height of the milk line significantly influenced the 1st WF/kg in the milking machine for sheep and goats ($p < 0.05$). The 1st WF increased as the milk jet diameter increased. The values increased from $2.91 \pm 0.05$ kg/min for the milk jet diameter of 2 mm to $5.51 \pm 0.06$ kg/min for the milk jet diameter of 4 mm in the sheep milking machine. The difference was highly significant ($p < 0.001$). Similar results were shown with the goat milking machine, with the values of the 1st WF/kg increasing from $3.03 \pm 0.11$ for the milk jet diameter of 2 mm to $5.08 \pm 0.16$ for the milk jet diameter of 4 mm ($p < 0.001$). Several lines of evidence indicate that the shape, size and canal diameter of the teat have a significant effect on the milk flow rate [14,30,31]. This influence is clearly noticeable between the breeds [32,33]. However, some breeds have a higher milk flow rate compared with other breeds even if the vacuum level, pulsation rate and pulsation ratio are the same.

**Table 2.** Some factors influencing the performance of the milking machine in sheep and goats.

| Parameter | Sheep | | Goats | |
|---|---|---|---|---|
| | N | LSMEANS ± SE | N | LSMEANS ± SE |
| | | Milk Jet (mm) | | |
| 2 | 57 | $2.91 \pm 0.05$ [a] | 54 | $3.03 \pm 0.11$ [a] |
| 2.5 | 57 | $3.85 \pm 0.05$ [b] | 54 | $3.63 \pm 0.11$ [b] |
| 3 | 54 | $4.82 \pm 0.05$ [c] | 42 | $4.47 \pm 0.13$ [c] |
| 4 | 42 | $5.51 \pm 0.06$ [d] | 24 | $5.08 \pm 0.16$ [d] |
| | | Vacuum (kPa) | | |
| 32 | 33 | $3.95 \pm 0.07$ [a] | 18 | $3.66 \pm 0.18$ [a] |
| 34 | 30 | $4.16 \pm 0.07$ [b] | 33 | $3.89 \pm 0.13$ [ae] |
| 36 | 57 | $4.26 \pm 0.05$ [b] | 36 | $4.05 \pm 0.13$ [be] |
| 38 | 30 | $4.44 \pm 0.07$ [c] | 42 | $4.27 \pm 0.12$ [cb] |
| 40 | 60 | $4.55 \pm 0.05$ [c] | 45 | $4.39 \pm 0.11$ [c] |
| | | Milk Line Height | | |
| Low Line | 110 | $4.51 \pm 0.04$ [a] | 90 | $4.41 \pm 0.06$ [a] |
| High Line | 100 | $4.03 \pm 0.04$ [b] | 84 | $3.70 \pm 0.16$ [b] |
| | | Pulsation Ratio | | |
| 50:50 | 105 | $4.23 \pm 0.05$ [a] | 95 | $3.92 \pm 0.23$ [a] |
| 60:40 | 105 | $4.32 \pm 0.03$ [a] | 79 | $4.07 \pm 0.19$ [a] |

Least square means in the same column and same variable with different superscripts a, b, c, d, e are statistically significantly different at $p < 0.05$.

In relation to the vacuum level, it was found that the 1st WF increased with rising vacuum level, and the values increased from $3.95 \pm 0.07$ kg/min at a vacuum of 32 kPa to $4.55 \pm 0.05$ kg/min at a vacuum of 40 kPa in the sheep milking machine. The difference was also significant ($p < 0.001$). A similar tendency was found when using the goat milking machine, and the 1st WF/kg was increased from $3.66 \pm 0.18$/kg at the vacuum level of 32 kPa to $4.39 \pm 0.11$/kg at the vacuum level of 40 kPa ($p < 0.001$). However, there were no significant differences between the vacuum levels of 38 and 40 kPa with respect to the 1st WF/kg in the sheep and goat milking machines ($p > 0.05$).

Previous studies have reported that the use of the milking machine in sheep when milking with a vacuum of 42 or 28 kPa did not show any significant differences in milk yield, while the milking time was increased from 0.94 to 1.10 min [21]. In this case, the low vacuum leads to low milk flow through the teat and thus extends the milking time. However, the authors found that delaying the milking time when using the 28 kPa vacuum level had inconsistent effects on the total milking time. The throughput of the milking system shows a slight difference between the two levels of vacuum: 333 sheep/h at 42 kPa vs. 321 at 28 kPa. In addition, a decrease in the milking vacuum (between 34 and 36 kPa) has been observed due to the decreasing mass of the milking clusters [34]. Previous studies have reported that the milk yield was satisfactory after using different vacuum levels in the milking machine, which shows that a low vacuum can be sufficient to completely empty the udder [21]. Reinemann [35] reported that a faster rate (120 pulsations/min) and a higher vacuum level (52 kPa) decreased the percentages of machine milk and total machine milk while increasing the percentage of machine-stripped milk and hand-stripped milk, after studying the effects of the vacuum level, pulsation rate and pulsation ratio on machine milking efficiency in local Greek goats. The milking time was also reduced when the vacuum level increased from 36 to 52 kPa and as pulsation ratios rose from 35:65 to 50:50 and 65:35.

The pulsation ratio also has an influence on the first minute of water flow in the sheep milking machine. However, the difference was not significant ($p > 0.05$), and the values were $4.23 \pm 0.05$ kg/min at the pulsation ratio of 50:50 and $4.32 \pm 0.03$ kg/min at the pulsation ratio of 60:40. Similar results were found when using goat milking machines (Table 2). In all experiments, the pulsation rate was always constant at 120 cycles/min for the sheep milking machine and 90 cycles/min for the goat milking machine. Previous studies have reported that most milked ewes had a higher pulsation rate (120 to 180 cycles/min), a low vacuum level (between 32 and 40 kPa) and a pulsation ratio of 50% [34,36]. In addition, a study on ewes (breed Manchega) showed that an increase in the pulsation rate from 120 to 180 cycles/min at 36 kPa vacuum and a pulsation ratio of 50:50 did not have any negative effects on the health of the udders or the condition of the teat ends [8]. Sinapis et al. [6] recommended that optimal conditions for machine milking of goats include a pulsation rate of 70–90 cycles/min, a vacuum level of 36–44 kPa and a pulsation ratio of 65:35. In the Alpine and Saanen breeds, it was found that a high pulsation rate of 90 to 120 cycles/min with a pulsation ratio of 60:40 reduced the milking time and increased the average milk flow rate, while a low pulsation rate of 60 cycles/min and pulsation ratio of 50:50 increased the milking time and decreased the average milk flow rate [37]. However, a high pulsation rate of 180 cycles/min and a low vacuum level are considered to be just as important as the optimal conditions for milking ewes and for udder health [38]. In addition, Reinemann [35] found that a pulsation rate of 180 cycles/min, compared to 120 cycles/min, had no negative impact on new intramammary infections and somatic cell count in the machine milking of ewes.

In this study, we used a pulsation rate of 120 cycles/min on the sheep milking machine as this could indicate better udder health. If we use 180 cycles/min with the new milking machine, as is also used in some sheep milking machines, then the rest phase in the cyclical pulsation rate is very short. Usually, this phase does not have to be shorter than 150 milliseconds. If the pulsation ratio of 60:40 is used in the milking machine, the rest phase is 133 milliseconds. In this case, teat health problems arise for a long period.

In the same way, the results of these tests clearly show that the milk line height affects the 1st WF/kg. The average 1st WF/kg was 4.51 ± 0.04 kg/min in the low-level milk line and 4.03 ± 0.04 kg/min in the high-level milk line in the milking machine for sheep, and the difference was significant ($p < 0.001$). Similar results were noted in the application of the goat milking machine. The results of Diaz et al. [3] showed that mid-level and low-level milking systems do not have a significant effect on milk production, milking time, the frequency of falling teat cups, somatic cell counts and milk composition, in Manchega ewes. Manzur et al. [39] reported that the milk fractioning (reduced machine milk and increased machine stripping) and average machine milk flow worsened slightly in the mid-level milk line in relation to the low-level milk line in goats; in contrast, no differences were observed in total milking time or teat thickness changes after milking.

As a result, in our study, optimum conditions for a new milking technique in goats with an air inlet teat cup appear to be a vacuum level of 36–38 kPa (low line) and 38–40 kPa (high line), a pulsation rate of 90 cycles/min and a pulsation ratio of 60:40, while in sheep, they appear to be a vacuum level of 35–36 kPa (low line) and 36–38 kPa (high line), a pulsation rate of 120 cycles/min and a pulsation ratio of 60:40 or 50:50.

### 3.2. Cyclical Vacuum Fluctuations in the Pulsation and Inner Chambers of the Teat Cup during the 1st Wf in the Sheep Milking Machine

The functional principle of a two-chamber milking machine is based on the periodic removal of milk from the udder using a vacuum. While there is a permanent vacuum within the teat liner and claw, the pulsation chamber is periodically subjected to either vacuum or atmospheric pressure by pulsation, which leads to an alternating opening and closing of the teat liner. When there is negative pressure both inside and outside the liner, the liner is open and the applied vacuum removes milk from the teat. If there is atmospheric pressure in the pulsation chamber, it collapses and exerts pressure on the teat tip, thereby interrupting the milk flow [40]. The rhythmic movement of the teat cup liner between open and closed positions causes the vacuum in the inner chamber of the teat cup to increase or decrease. This phenomenon is called cyclic vacuum fluctuation during milking and can be measured with the aid of a Vacuscope device. If we look at the diagrams in Figures 1–4, it can be seen that the high milk line clearly influences the vacuum level in the interior of the teat cup liner during the suction phase. The vacuum level in the inner chamber was low, regardless of the milk jet, when the milk line was high. In contrast, the vacuum level in the inner chamber was equal to or slightly higher than the vacuum level in the pulsation chamber when the milk line was low. This change in the vacuum level in the inner chamber of the teat cup is independent of the vacuum set on the milking machine. In addition, the vacuum level of 34 kPa in the milking machine seems to be sufficient, but when the diameter of the milk jet (or streak canal in the teat) is above 2.5 mm, the milking machine is loaded (sometimes falling down), especially if the milk line is high (Figure 1). We therefore recommend using a vacuum level between 35 and 36 kPa for a low-level milk line and 36 and 38 kPa for a high-level milk line (Figures 2 and 3). Based on our observations, a vacuum level higher than 38 kPa is not required in the sheep milking machine with a pulsation rate of 120 cycles/min and a pulsation ratio of 60:40 (Figure 4). Conversely, a high vacuum level has an impact on udder health and milk quality. Caria et al. [41] clearly emphasized that a high vacuum level can lead to congestion of the teat, the formation of oedema, an increase in residual milk and an increase in the somatic cells in the produced milk. Moreover, Caria et al. [21] found that a low vacuum level alters the kinetics of milk removal, the machine-on time and thus the throughput of the milking system in sheep and goats. The results show that the 1st WF performance was satisfactory at the vacuum level tested (34 to 38 kPa). This means that a low vacuum level can be sufficient to completely empty the udder when the milking machine is used directly on the sheep. Skapetas et al. [17] recommended a scheme for sheep with a vacuum level of 34–38 kPa, a pulsation rate of 150 cycles/min and a pulsation ratio of 50:50; the basis for these recommendations was that the local Greek sheep breeds can be milked with a lower vacuum level as the vacuum required for the

teat opening is lower [42]. In the same way, Billon et al. [34] reported that the working parameters (vacuum level, pulsation rate and pulsation ratio) of the milking machines for dairy sheep have a direct influence on the milkability of the animals and the total milking time of the herd. Finally, the lowest vacuum level should be used to milk in order not to excessively extend the milking time and to take into account the maintenance of animal welfare in dairy husbandry [21].

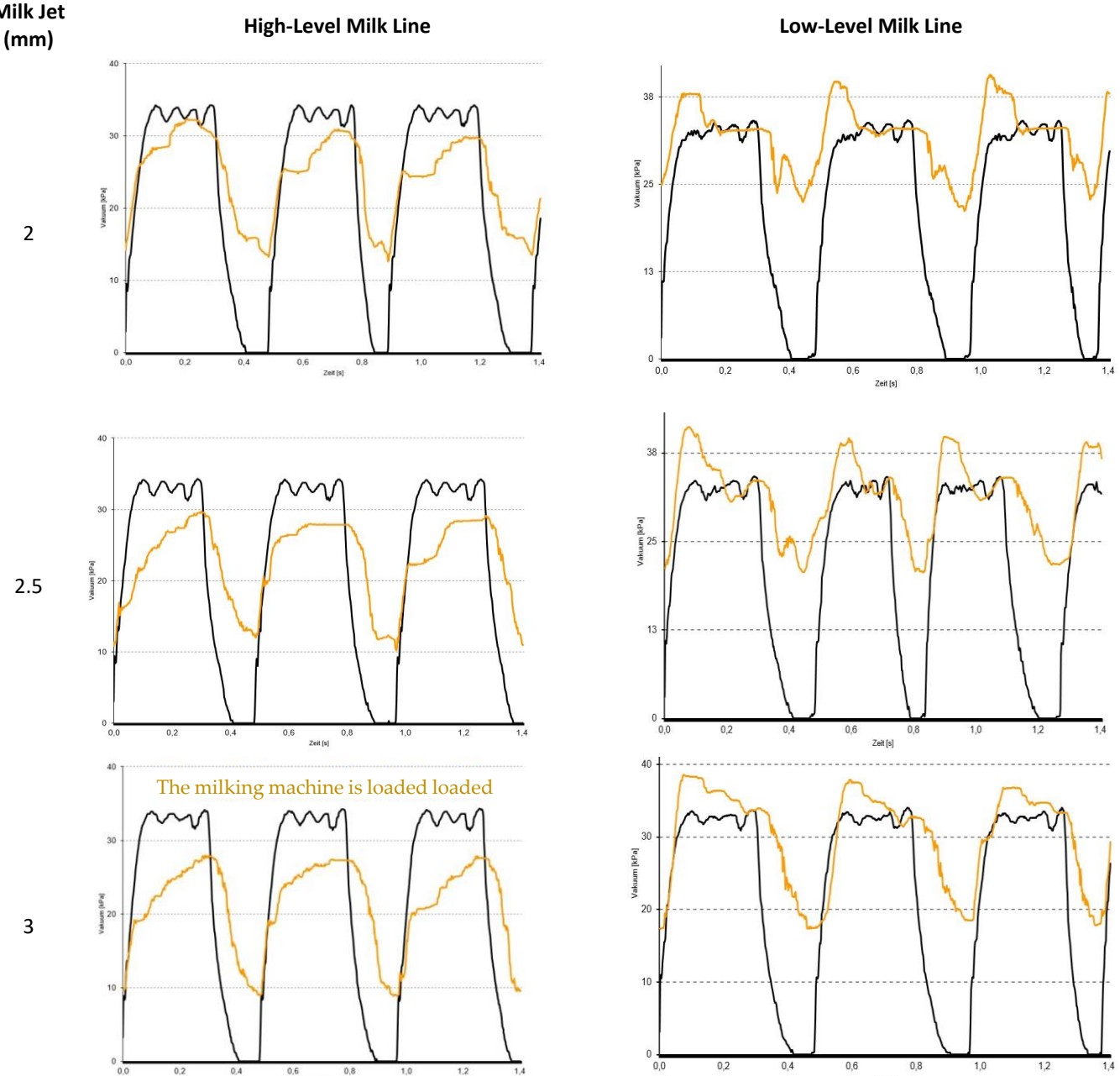

**Figure 1.** Cyclical vacuum fluctuations in the pulsation and inner chambers of the teat cup during milking the 1st WF in the sheep milking machine with the following settings: 34 vacuum level (kPa), 120 pulsation rate (cycles/min) and 60:40 pulsation ratio.

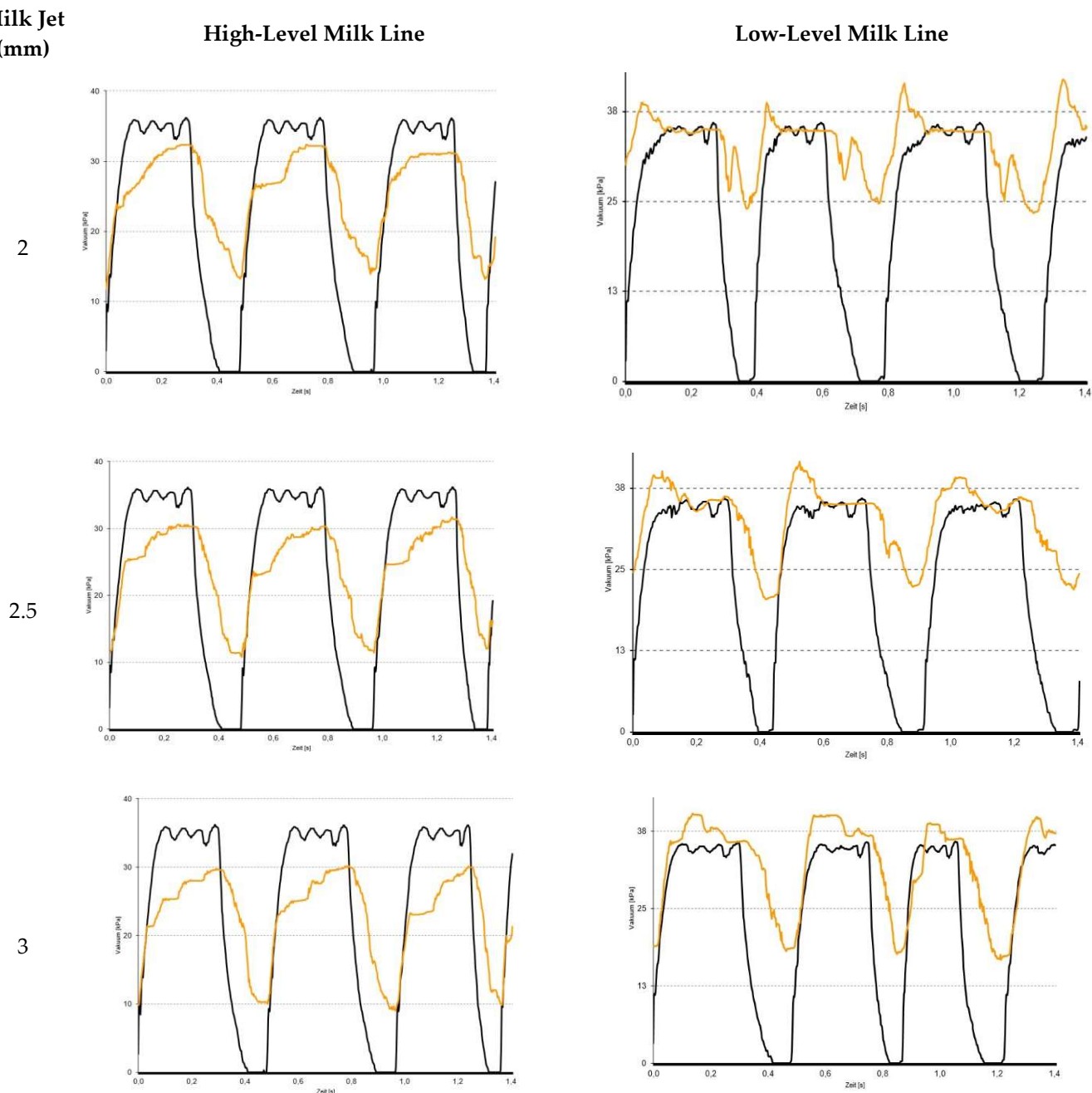

**Figure 2.** Cyclical vacuum fluctuations in the pulsation and inner chambers of the teat cup during the 1st WF in the sheep milking machine with the following settings: 36 vacuum level (kPa), 120 pulsation rate (cycles/min) and 60:40 pulsation ratio.

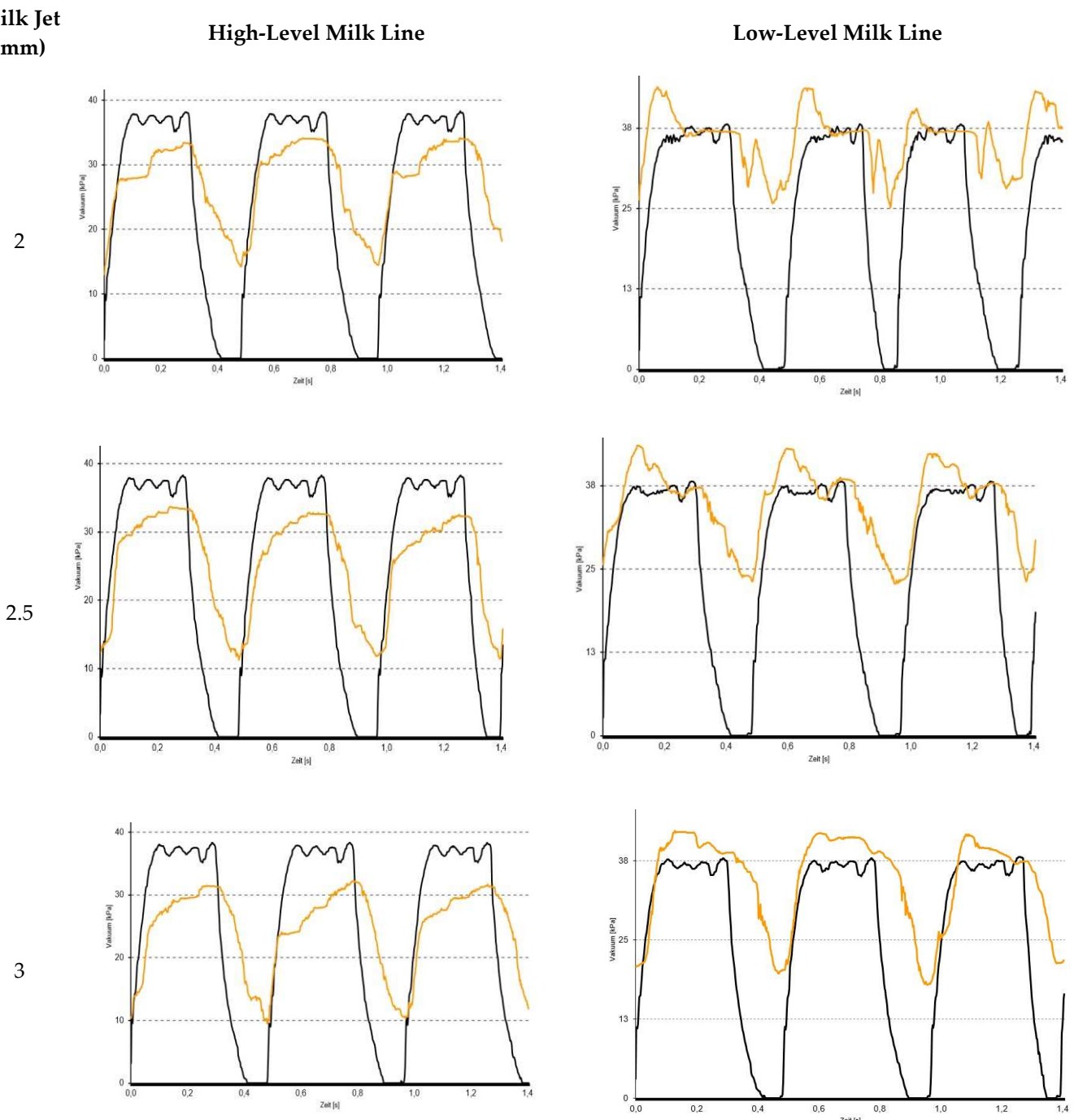

**Figure 3.** Cyclical vacuum fluctuations in the pulsation and inner chambers of the teat cup during milking the 1st WF in the sheep milking machine with the following settings: 38 vacuum level (kPa), 120 pulsation rate (cycles/min) and 60:40 pulsation ratio.

**Milk Jet (mm)**

**High-Level Milk Line**                                    **Low-Level Milk Line**

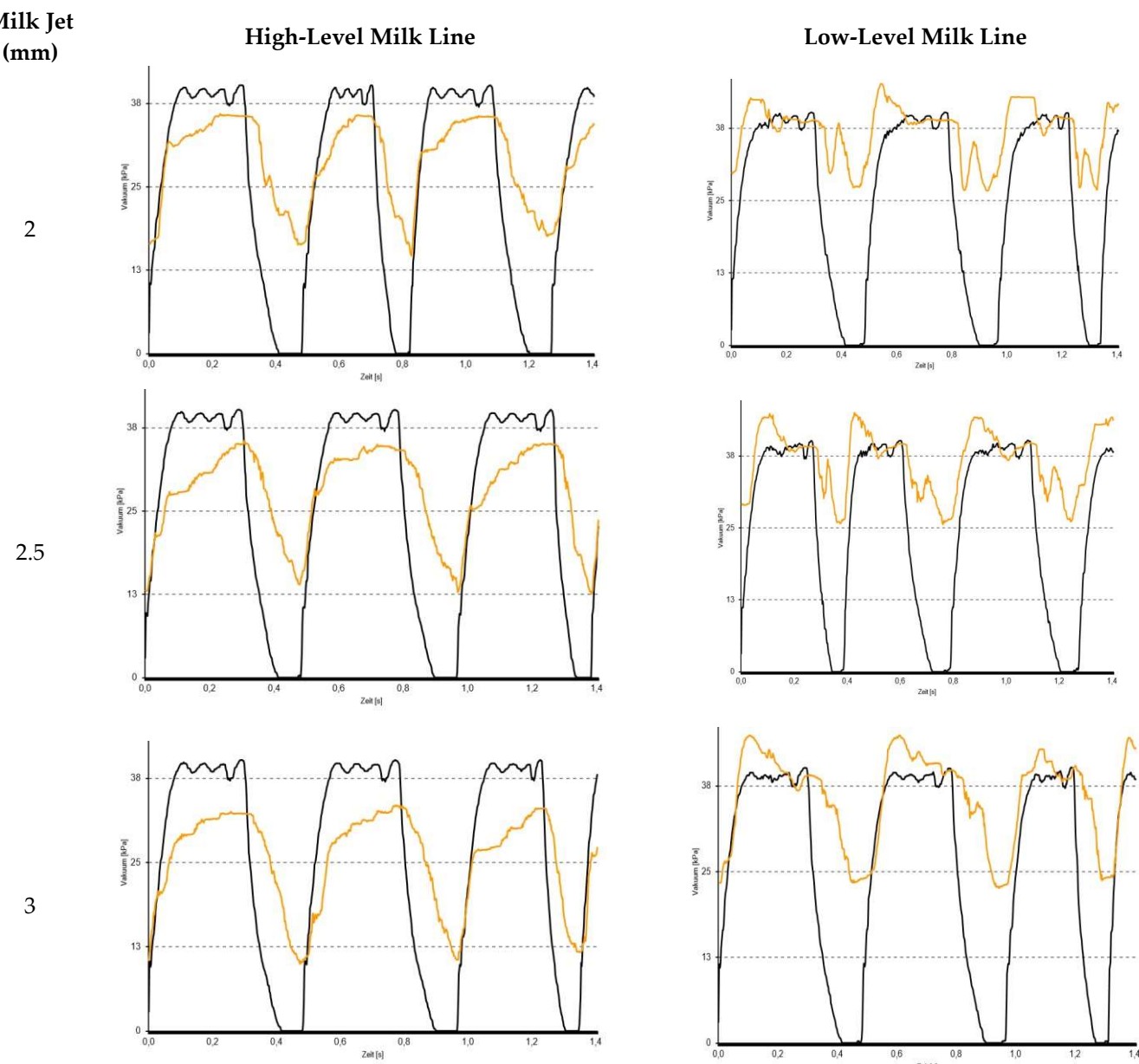

2.5

**Figure 4.** Cyclical vacuum fluctuations in the pulsation and inner chambers of the teat cup during milking the 1st WF in the sheep milking machine with the following settings: 40 vacuum level (kPa), 120 pulsation rate (cycles/min) and 60:40 pulsation ratio.

*3.3. Cyclical Vacuum Fluctuations in the Pulsation and Inner Chambers of the Teat Cup during the 1st Wf in the Goat Milking Machine*

The diagrams in Figures 5–8 show the cyclical vacuum fluctuations in the pulsation chamber and in the interior of the teat cups during the 1st WF in the goat milking machine. Although the vacuum level in the pulsation chamber was increased up to 40 kPa, the vacuum level in the inner chamber of the teat cups remained lower during the milking process in the high-level milk line as compared to the low-level milk line. Similar results were noted by other authors [3,39,43]. Additionally, Manzur et al. [39] found that the mean value of the vacuum during milk flow in goats decreased significantly in the middle milk line, while only slight fluctuations were observed in the lower milk line, the values being 33.6 and 36.5 kPa in the middle and lower milk lines, respectively. To explain this, it can be noted that an accumulation of milk in the claw and in the milk line, which has to be transported by the milk line vacuum, impairs the free air flow and leads to a reduction

in the vacuum in the inner chamber of the teat cups. This means that the upward milk transport requires considerable force (vacuum), meaning that the vacuum in the inner chamber of the teat cup decreases. In the 1970s, Le Du [44] described the situation of the hydrostatic pressure exerted by the air–milk mixture when it climbs the long milk line, and the higher the milk flow, the steeper this drop becomes. The general decrease in the vacuum level during the milking process depends on the milk flow rate and the height of the milk line. However, this problem did not occur with a low-level milk line condition, regardless of the vacuum level in the pulsation chamber of the teat cup. Conversely, the vacuum level in the inner chamber of the teat cup in the low-level milk line, in most cases, increased above the vacuum in the pulsation chamber (Figures 5–8).

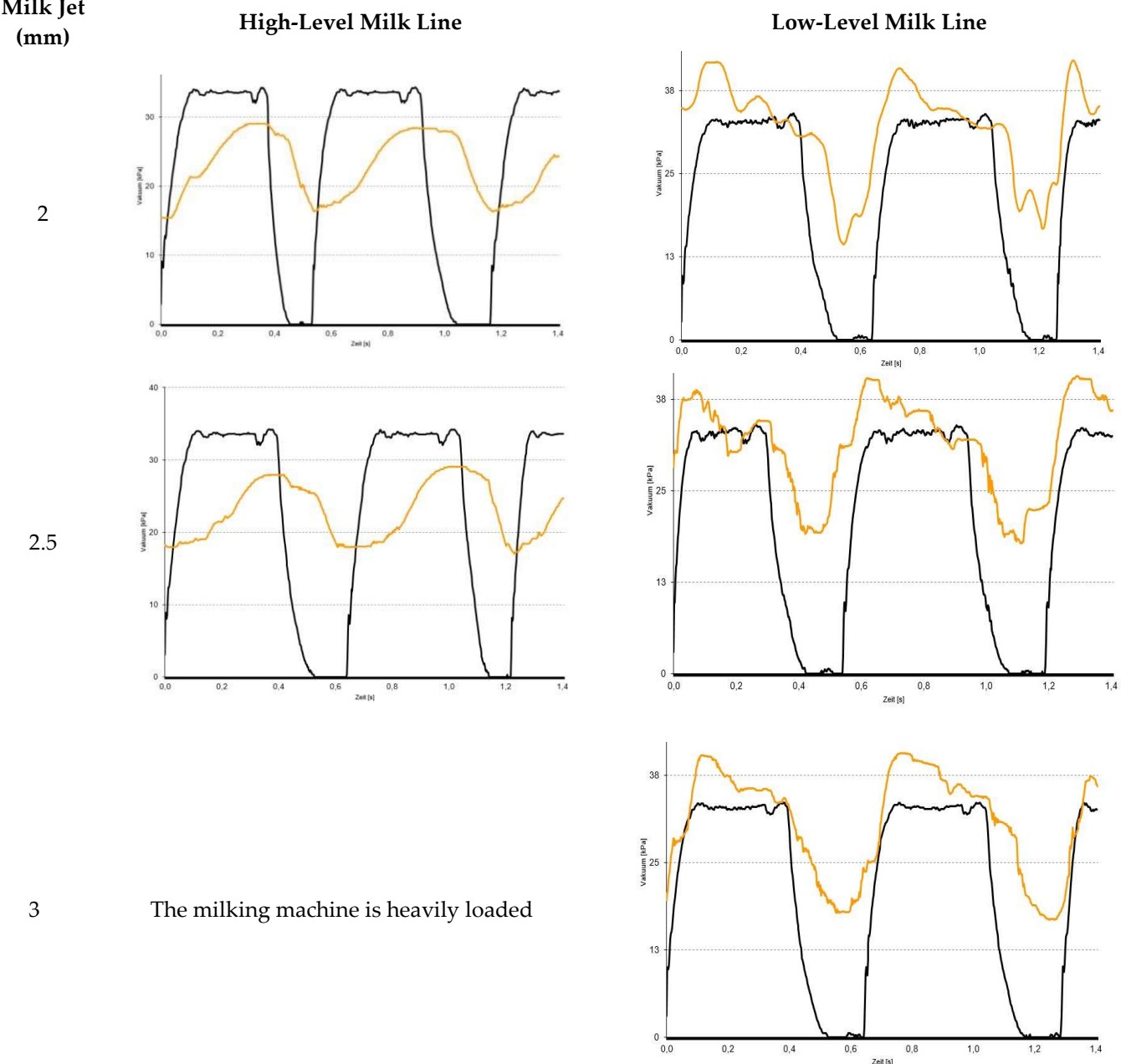

**Figure 5.** Cyclical vacuum fluctuations in the pulsation and inner chambers of the teat cup during the 1st WF in the goat milking machine, with the following settings: 34 vacuum level (kPa), 90 pulsation rate (cycles/min) and 60:40 pulsation ratio.

**Milk Jet (mm)**

**High-Level Milk Line**  **Low-Level Milk Line**

2.5

The milking machine is loaded

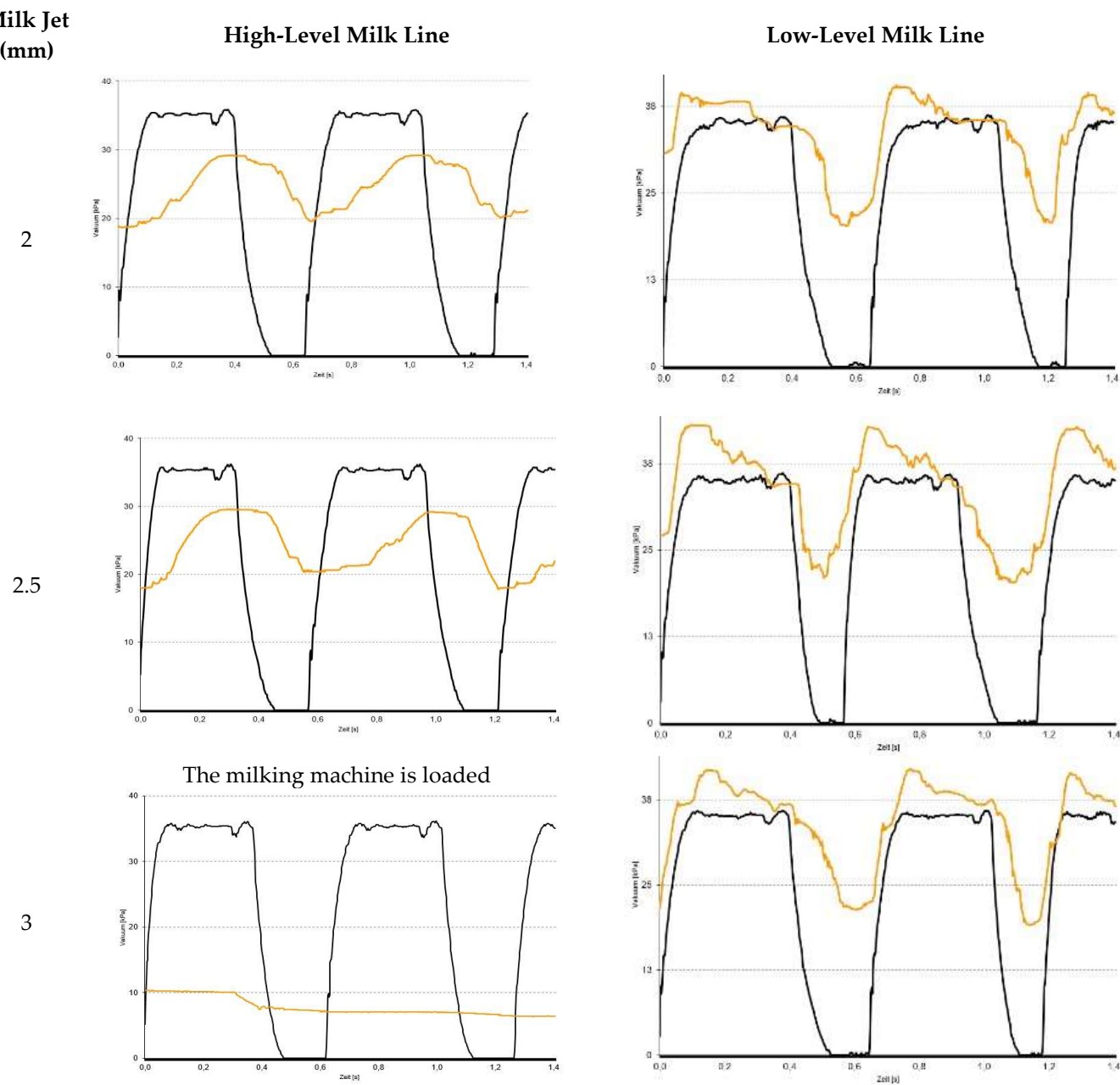

**Figure 6.** Cyclical vacuum fluctuations in the pulsation and inner chambers of the teat cup during the 1st WF in the goat milking machine, with the following settings: 36 vacuum level (kPa), 90 pulsation rate (cycles/min) and 60:40 pulsation ratio.

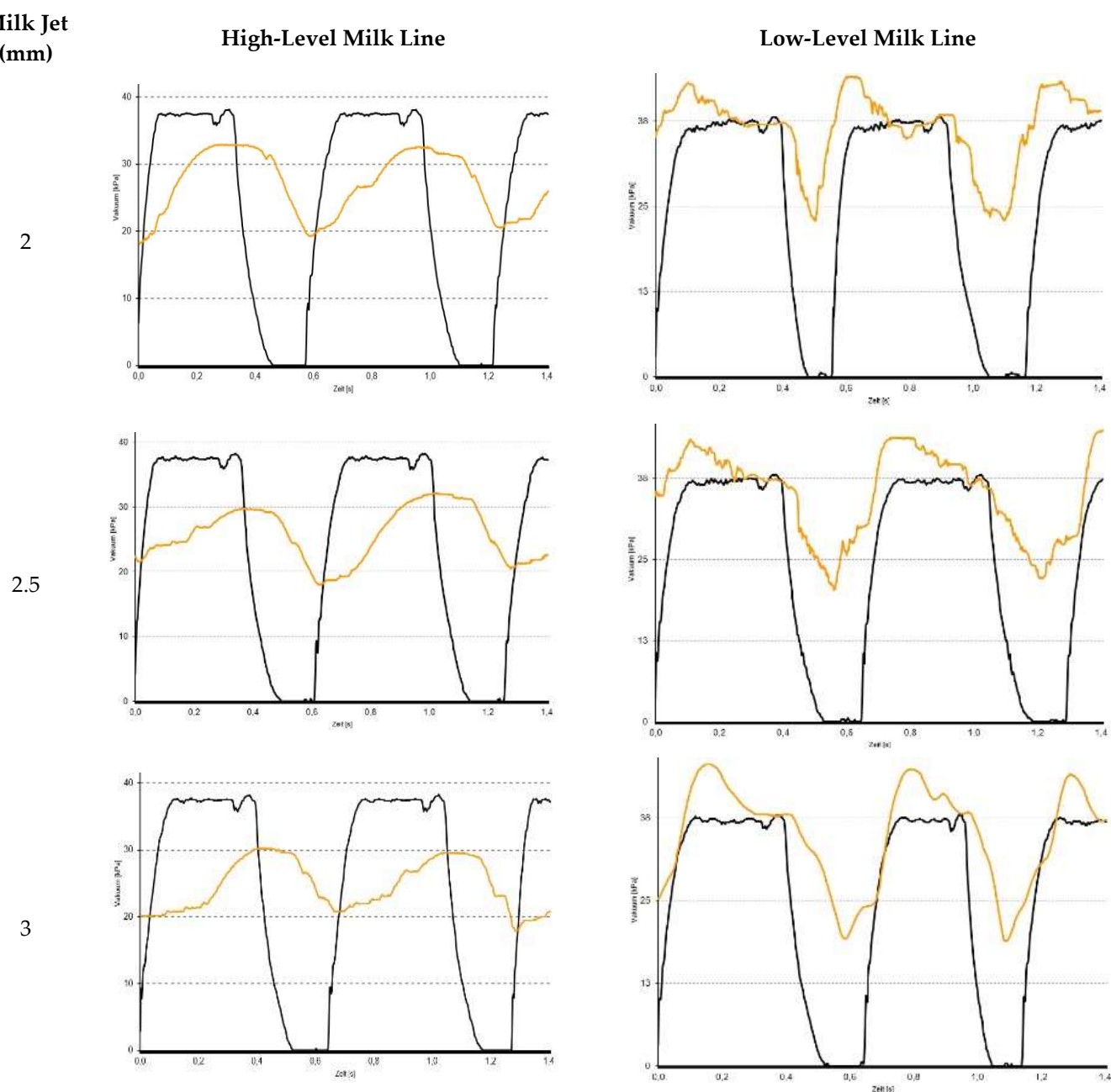

**Figure 7.** Cyclical vacuum fluctuations in the pulsation and inner chambers of the teat cup during the 1st WF in the goat milking machine, with the following settings: 38 vacuum level (kPa), 90 pulsation rate (cycles/min) and 60:40 pulsation ratio.

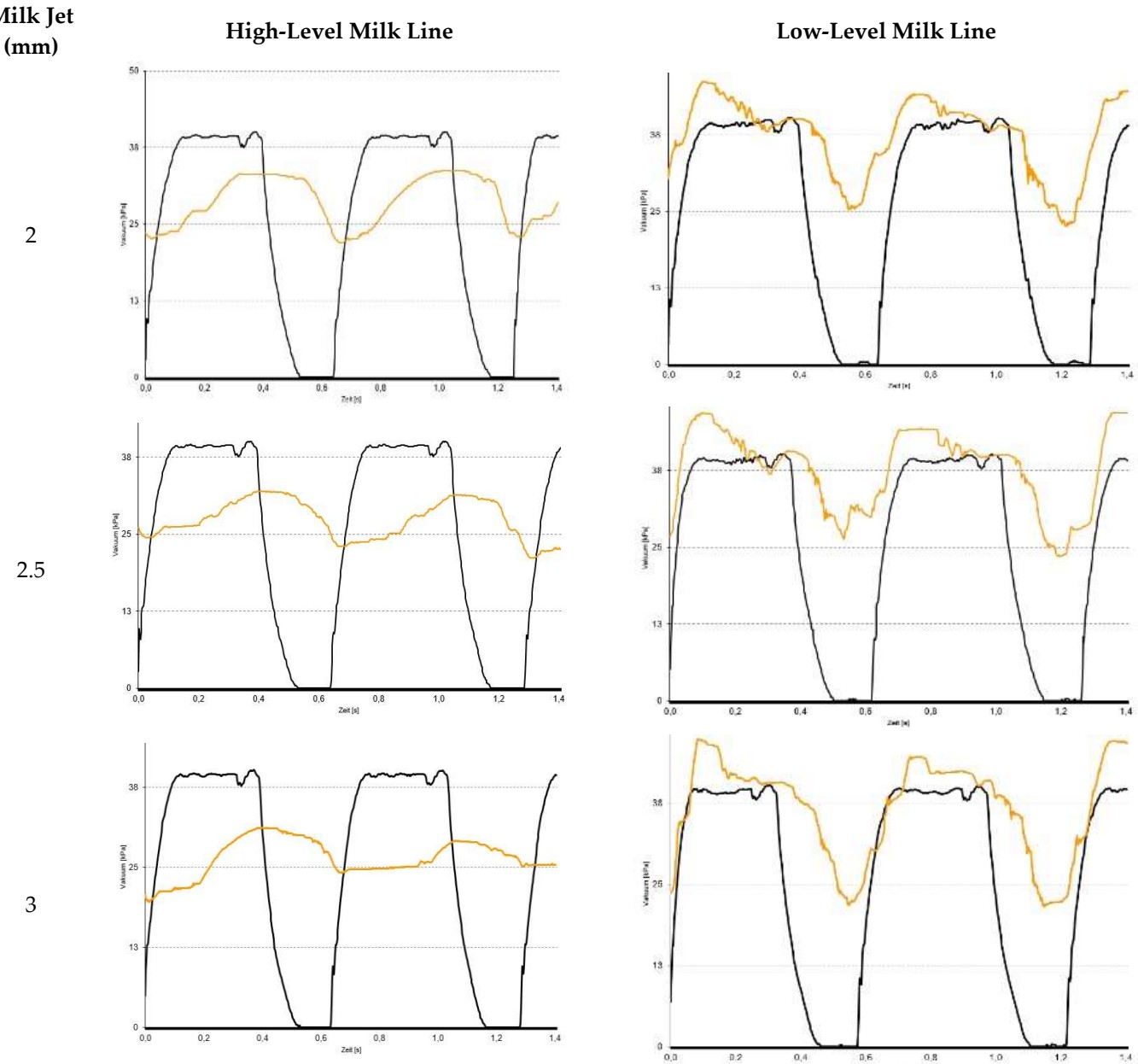

**Figure 8.** Cyclical vacuum fluctuations in the pulsation and inner chamber of the teat cup during the 1st WF in the goat milking machine, with the following settings: 40 vacuum level (kPa), 90 pulsation rate (cycles/min) and 60:40 pulsation ratio.

In addition, if the milk flow has been increased by increasing the milk jet to 3 mm and, at the same time, the vacuum levels in the pulsation chamber have remained at a low 34 or 36 kPa, then the milking machine cannot function sufficiently in the high-level milk line (Figures 5 and 6). This problem did not occur when the milk line was low. In this case, a higher vacuum above 36 kPa must be used in the milking machine (in the pulsation chamber) in order to maintain the efficiency of the milking machine and to pump the milk into the upper milk line, as shown in Figures 5 and 6. It is noteworthy that irregular and cyclical fluctuations in the vacuum level of the milking machine cause the milk outlet at the teat end to kick back with a low force or even to pump milk back into the teat canal, meaning that pathogens can probably penetrate the udder [36]. This situation occurs during mechanical stripping due to the air penetrating into the milking machine, resulting in a

kickback of the milk, which also occurs when the milking machine is removed from the udder without switching off the vacuum.

With the above results in mind, we therefore recommend using a vacuum level between 36 and 38 kPa for a low-level milk line and 38 and 40 kPa for a high-level milk line.

## 4. Conclusions

In summary, it was found that a vacuum level of 36–38 kPa (low line) and 38–40 kPa (high line), a pulsation rate of 90 cycles/min and a pulsation ratio of 60:40 are necessary for the goat milking machine with an air inlet teat cup, while sheep milking machines require a vacuum level of 34–36 kPa (low line) and 36–38 kPa (high line), a pulsation rate of 120 cycles/min and a pulsation ratio of 60:40.

In addition, silicone liners that are adapted to the morphological characteristics of each species (sheep or goat) must be used to ensure the benefits of pulsation at the end of the teat.

**Funding:** There is no additional funding for this research.

**Institutional Review Board Statement:** The study was conducted in accordance with German and European legislation (directive 2010/63/EU).

**Informed Consent Statement:** Not applicable.

**Data Availability Statement:** All data used to support the findings of this study are included within the article.

**Conflicts of Interest:** The author declares no conflict of interest in this work.

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
