# Peer review of "Laboratory Tests to Optimize the Milking Machine Settings with Air Inlet Teat Cups for Sheep and Goats"

_2624-862X, doi:10.3390/dairy3010003_

Round 1
Reviewer 1 Report
I have an initial serious criticism about this paper. Testing with animals and control testing (without air inlet teat cup) were not included in the present study, and methodology was poorly defined. Thus, dependent variables studied, differences in sheep/goat milking machines, devices to obtain multiple milk jets, pump air capacity, manual reserve, milking units, etc. were not defined. The factors included is the study, particularly the pulsation rate, were not optimized: 120 cycles/min in goat and 180 cycles/min in sheep would have been of interest. Vacuum fluctuations should be quantified, etc. Therefore this paper does not add new knowledge, as designed, and results are inconclusive. In addition, the 1st WF/kg variable must be clearly defined. Is it maximum milk flow, average flow, milk yield milked in the 1st minute...?
Author Response
- reviewer: Thank you for your comments.
I have an initial serious criticism about this paper.
When developing a new milking machine, a laboratory test is first required and is then tested directly on the animals. After successful tests in the laboratory and in practice, the milking machines are ready for the market.
Testing with animals and control testing (without air inlet teat cup) were not included in the present study, the methodology was poorly defined.
Our work was only shown in the laboratory tests for milking machines for sheep and goats. In this case, no comparison with a control group is necessary.
Thus, depend variables studied, differences in sheep/ goat milking machines, devices to obtain multiple milk jets, pump air capacity, manual reserve, milking units, etc. were not defined.
All of the above information is contained in the improved manuscript.
The factors included is the study, particularly the pulsation rate, were not optimized: 120 cycles/min in goat and 180 cycles/min in sheep would have been if interest.
After intensive literature research in this area, it has been shown that a pulsation rate of 90 cycles / minute for goats and 120 cycles / minute for sheep provides optimal results for the milking machine.
Vacuum fluctuations should be quantified, etc.
Yes, it is possible, but not necessary, as none will give better results than those shown in the figures.
Therefore, this paper does not add new knowledge, as designed, and results are inconclusive.
I accept your opinion, but not all problems can be solved in one article as many farmers have practical problems with existing milking machines and want new milking technology to solve the problems with milk removal in goats and sheep.
In addition, the 1st WF/kg variable must be clearly defined. is it maximum milk flow, average flow, milk yield milked in the 1st minute...?
Yes, you are right and you can see the answer in the improved manuscript.
Reviewer 2 Report
Dear author,
you did some interesting lab research. As you mentioned, the quality of machine milking of goats and sheep is observed through the number of somatic cells in milk. You designed the experiment well. You have studied a large number of references. However, the actual research on animals is often quite different from that in the laboratory. You have stated that this will be your next task.
The hypothesis of your work is not completely clear to me. What is the contribution of your innovation to the milking machine, which consequently influenced the results obtained in your research? Do you think that a milking machine would have completely different results without your innovation? I would ask you for clarification.
Also, Tables 2-5 are just vacuum and pulsation diagrams. No impact is seen here. Such data are part of the research methodology and are placed in the chapter Material and Methods of work. In the Results and Discussion section, more information is expected on certain impacts that you listed in Section 2.4. Statistical analysis. The article has too little research data.
Based on the above, the paper falls into the category of Preliminary Communication.
With certain answers and clarifications.
Author Response
- reviewer: Thank you for your comments.
The hypothesis of your work is not completely clear to me.
Yes, you are right and you can see the answer in the improved manuscript.
What is the contribution of your innovation to the milking machine, which consequently influenced the results obtained in your research?
We have developed a new milking machine with an air inlet on the teat cup for sheep and goats, which are still in the test phase. This work demonstrated only the laboratory tests. Then the practice is carried out directly on the animals.
Do you think that a milking machine would have completely different results without your innovation? I would ask you for clarification.
Yes of course, because our milking machine has special features that the other milking machine does not have. The new milking machine has the following features:
- It has a special silicone teat liner.
- It works with a low vacuum level.
- The system is equipped with a periodic air inlet into the teat cup.
Also, Tables 2-5 are just vacuum and pulsation diagrams. No impact is seen here.
Much can be seen in Tables 2-5 (Illustrations 1 to 8):
- Whether a change in the vacuum level can be seen in the inner chamber of the teat cup than in the pulse chamber. Since in many milking machines the vacuum in the inner chamber of the teat cup cannot reach the vacuum in the pulsation chamber, problems can arise during the suction phase when removing the milk.
- Whether the vacuum level in the inner chamber of the teat cup changes when the milk jet (diameter of the teat) changes.
- The difference in the vacuum level in the inner chamber of the teat cups between the high-level milk line and the low-level milk line can also be clearly seen. If the vacuum in the inner chamber of the teat cup does not reach a certain value, then teat cup can fall to the ground.
Such data are part of the research methodology and are placed in the chapter Material and Methods of work.
Such diagrams are results and are correctly inserted into the results part. In my opinion.
In the results and discussion section, more information is expected on certain impacts that you listed in Section 2.4 Statistical analysis. The article has too little research data.
We received a lot of data and results from the work. We have planned some results for publication in other articles.
Based on the above, the paper falls into the category of preliminary communication.
With certain answers and clarifications.
I tried to explain your question clearly so that you can better understand the paper. Perhaps some questions will be clarified in the improved manuscript.
Reviewer 3 Report
The manuscript has not been formatted according to the standards of the journal.
Introduction
The hypothesis of the author is not presented clearly.
The objectives of the study should be defined.
Materials and methods
2.2. The various parameters should be included in a table, not in the text.
2.3. Please define clearly all the parameters assessed.
2.4. What statistical model did you use? How you confirmed that results were normal?
Results
Please use exact p values everywhere in the manuscript, not dichotomous approach.
Please include figures with graphical presentations of findings.
Tables 2 to 5 are not tables, they are figures.
Did you test animals for development of mastitis? If no, why?
The manuscript has merit and can be accepted after submission of a corrected and improved version. The points mentioned above should be carefully corrected.
The revised manuscript needs to be evaluated again.
Author Response
- reviewer: Thank you for your comments.
The manuscript has not been formatted according to the standards of the journal.
I did exactly what the journal asked. That said, I did exactly how an article was published in the journal.
Introduction, the hypothesis of the author is not presented clearly. The objectives of the study should be defined.
Our hypothesis was to develop a new milking machine for sheep and goats that would guarantee high milk yield and quality while maintaining udder health.
Our objective is to optimize the vacuum level, the pulsator rate and the ratio in the new milking machine in the higher and lower milk lines.
Material and methods,
2.2. The various parameters should be included in a table, not in the text.
Please see the improved manuscript
2.3. please define clearly all the parameters assessed
Please see the improved manuscript.
What statistical model did you use? How you confirmed that results were normal?
See the improved manuscript. Yes, the data was normally distributed. Usually everyone has to check the distribution of the data first, whether it is normally distributed or not. This can easily be checked with the SAS program.
Results, please use exact p values everywhere in the manuscript, not dichotomous approach.
Please see the improved manuscript
please include figures with graphical presentations of findings
Please see the improved manuscript.
Tables 2 to 5 are not tables, they are figures.
See the improved manuscript
Did you test animals for development of mastitis? If no, why?
The manuscript contains information about laboratory tests, not animals. This phase with animals will be carried out later.
Round 2
Reviewer 1 Report
Reviewer suggestions have been neglected. Title must be changed. This paper is not an optimization of the milking machine, but a study of laboratory test performance of milking machine. On the other hand, the paper should be drastically shortened. Please, delete Scheme 1 and 2 and summarize Illustrations 1 to 8 in one o two (sheep and goat) Tables. In my opinion this paper has little interest because it should be completed with field tests. I recommend the publications as a Short communication after thorough revision concernig presentation of results.
I think choice of parameters was not optimized. 180 cycles/min is a pulsation rate widely used in dairy sheep. See, for example, Fernandez et al. (1999): Machine milking parameters for the Manchega sheep breed, In: Milking and milk production of dairy sheep and goats. Proceedings 6th Int. Symp. Milking of Small Ruminants, Athens, Greece, 1998, EAAP Publication 95: 233-238; or Murgia and Pazzona (1999): Comparison among six milk claws for sheep milking, In: EAAP Publication 95: 245-247; or Gonzalo et al. (2019): Bulk tank somatic cell count and total bacterial count are affected by target practices and milking machine features in dairy sheep flocks in Castilla y Leon region, Spain. Small Rumin. Res. 178: 22-29; etc. A discussion about that is needed
Other additional comments:
L97-98: Please, change to: the amount of the water received (milked) in kg in the first minute after teat attachment.
Page 6: Legend of Table 2: Variables (and units) must be clearly indicated.
Illustrations 1 to 8: Vacuum fluctuations should be quantified and summarized in one o two Tables. These results must be drastically shortened.
L391: Fernandez instead of Fernandz
Author Response
Thank you for your comments
Reviewer suggestions have been neglected.
Perhaps you have not been properly informed. You can show in the first manuscript and find the difference between the improved manuscript and the first manuscript.
Title must be changed. This paper is not an optimization of the milking machine, but a study of laboratory test performance of milking machine.
is ready, see the revised manuscript
On the other hand, the paper should be drastically shortened. Please, delete Scheme 1 and 2 and summarize Illustrations 1 to 8 in one two (sheep and goat) Tables.
I accept your views, but the other reviewers have suggested more details and they agreed to these new suggestions.
In my opinion this paper has little interest because it should be completed with field tests. I recommend the publications as a short communication after thorough revision concerning presentation of results.
- I accept your opinion, but every milking machine must first be examined in the laboratory before being used in the field.
- Usually, the companies that make milking machines don't show the lab tests. They want this investigation to remain a secret, but we will fix everything. Therefore, laboratory tests are of paramount importance in the manufacture of new milking machines and before they are used.
I think choice of parameters was not optimized. 180 cycles/min is a pulsation rate widely used in dairy sheep. See, for example, Fernandez et al. (1999): Machine milking parameters for the Manchega sheep breed, In: Milking and milk production of dairy sheep and goats. Proceedings 6th Int. Symp. Milking of Small Ruminants, Athens, Greece, 1998, EAAP Publication 95: 233-238; or Murgia and Pazzona (1999): Comparison among six milk claws for sheep milking, In: EAAP Publication 95: 245-247; or Gonzalo et al. (2019): Bulk tank somatic cell count and total bacterial count are affected by target practices and milking machine features in dairy sheep flocks in Castilla y Leon region, Spain. Small Rumin. Res. 178: 22-29; etc. A discussion about that is needed.
We used the above references in the discussion, but you may not have noticed. We used 120 cycles/min as the pulsation rate in the sheep milking machine for the following reasons:
- We are using a new sheep milking machine and the results presented are laboratory results. This means that when the field studies are complete, we can determine the correct pulsation cycle.
- If we use 180 cycles/min as pulsation rate on the new milking machines. Then the rest phase is very short and normally this phase does not have to be shorter than 150 milliseconds. If a pulsation ratio of 60:40 is used in the milking machine, the rest phase is 133 milliseconds. In this case, teat health problems arise for a long period.
L97-98: Please, change to: the amount of the water received (milked) in kg in the first minute after teat attachment.
is ready, see the revised manuscript
Page 6: Legend of Table 2: Variables (and units) must be clearly indicated.
is ready, see the revised manuscript
Illustrations 1 to 8: Vacuum fluctuations should be quantified and summarized in one or two Tables. These results must be drastically shortened.
If vacuum fluctuations are to be quantified in Illustration 1 to 8 and summarized in one or two tables, no other results will be obtained. Conversely, many scientists see the results clearly in this form and see the difference immediately, as if the results were presented in a table. This is my opinion.
L391: Fernandez instead of Fernandz
is ready, see the revised manuscript
Reviewer 2 Report
Dear author,
in your article you took into account my remarks that I indicated to you during the review. I think that with these additions you have significantly improved the content of the article. In the Introduction you have clearly written the hypothesis of the paper. The chapter Material and Methods, ie Experimental Design, has been significantly expanded with a new scheme and Statistical Analysis. In the chapter Results and Discussion you have added a new table of individual influences on the investigated milking parameters. You have clarified illustrations 1-8 with new information.
Based on the above, the paper belongs to the scientific category, and I suggest an article for publication.
Author Response
Thank you for your comments and positive answer.
Reviewer 3 Report
Thank you for revising the manuscript.
Author Response

(The authors gave the same response as above.)
